# The Impact of Gut Microbiota on the Development of Anxiety Symptoms—A Narrative Review

**DOI:** 10.3390/nu17060933

**Published:** 2025-03-07

**Authors:** Kamil Nikel, Michał Stojko, Joanna Smolarczyk, Magdalena Piegza

**Affiliations:** 1Students Scientific Association, Department of Psychiatry, Faculty of Medical Sciences in Zabrze, Medical University of Silesia in Katowice, 40-055 Katowice, Poland; s83148@365.sum.edu.pl; 2Department of Psychoprophylaxis, Medical University of Silesia in Katowice, 40-055 Katowice, Poland; joanna.smolarczyk@sum.edu.pl; 3Department of Psychiatry, Faculty of Medical Sciences in Zabrze, Medical University of Silesia in Katowice, 40-055 Katowice, Poland; mpiegza@sum.edu.pl

**Keywords:** microbiota, anxiety disorders, mental health

## Abstract

The gut microbiota plays a key role in mental health, with growing evidence linking its composition to anxiety and depressive disorders. Research on this topic has expanded significantly in recent years. This review explores alterations in the gut microbiota of individuals with anxiety disorders and examines the potential therapeutic effects of probiotics. **Background/Objectives**: This review aims to analyze the alterations in gut microbiota composition in individuals with anxiety disorders and evaluate the potential therapeutic effects of probiotics in mitigating symptoms. By examining recent research, this study seeks to highlight the gut–brain connection and its implications for mental health interventions. **Materials and Methods**: A literature search was conducted in PubMed, Embase, CINAHL, and Google Scholar, focusing on studies investigating the relationship between gut microbiota and anxiety disorders, as well as the impact of probiotics on symptom severity. **Results**: The reviewed studies suggest that individuals with anxiety disorders often exhibit gut microbiota alterations, including reduced microbial diversity and a lower abundance of short-chain fatty acid-producing bacteria. Additionally, probiotics, particularly those from the *Lactobacillus genus*, may help alleviate anxiety symptoms by modulating gut microbiota composition. **Conclusions**: Gut dysbiosis appears to be closely linked to anxiety disorders, and probiotic interventions could represent a promising therapeutic avenue. However, further research is needed to clarify underlying mechanisms and optimize treatment strategies.

## 1. Introduction

In recent years, there has been a rapid increase in scientific publications concerning the role of gut microbiota in shaping human health and the relationship between gut dysbiosis and various diseases. It has long been known that the role of gut microbiota is not limited to digestion and absorption of nutrients. The microbiota is composed primarily of bacteria, but also includes viruses, fungi, and archaea, which interact with each other and with the host organism. Emerging research suggests that these microorganisms may also play a role in mental health, including anxiety disorders. Certain viruses and fungi can influence immune responses and inflammation, which are linked to anxiety symptoms, while archaea may impact neurotransmitter metabolism and gut–brain signaling, potentially affecting mood regulation [1,2,3]. The gut microbiota is influenced by several factors, including genetics, diet, age, and geographical location, all of which may affect its role in anxiety disorders. Genetic factors shape microbiota diversity and stability, potentially impacting neurotransmitter production and immune function, both of which are linked to anxiety. Diet is another key determinant—fiber-rich and fermented foods support beneficial bacteria, while highly processed diets can lead to imbalances. Microbiota composition also changes with age, with a decline in beneficial strains potentially increasing vulnerability to anxiety. Additionally, regional differences in diet, sanitation, and environmental microbial exposure contribute to variations in gut bacteria, which may play a role in mental health. Recognizing these influences could help develop personalized strategies for managing anxiety through microbiota-targeted interventions [4,5].

The bacteria within it perform many functions, and their proper quantitative and qualitative structure (referred to as eubiosis) supports the homeostasis of the entire organism, influencing immunity, metabolism, and the synthesis of various chemical compounds, such as serotonin and other neurotransmitter precursors such as dopamine, GABA (gamma-aminobutyric acid), and acetylcholine. Since some gut bacteria are pathogenic, beneficial species should predominate to support intestinal health [4]. A positive correlation between quantitative and qualitative disturbances in gut microbiota and the occurrence of anxiety disorders in patients has also been observed, and recent reports confirm this information. Anxiety disorders are among the most common mental health problems, affecting more than 4% of the population (2023) and are often diagnosed in primary care settings [6,7].

There are many theories that explain the influence of gut microbiota on neurophysiological processes. Among other things, attention should be paid to the immune and metabolic functions of microbiota. It is responsible for providing essential vitamins and nutrients and can recognize lipopolysaccharide (LPS) molecules present in pathogens through Toll-like receptors (TLRs), triggering an inflammatory response through mediators and cytokines. Disruptions in these processes can lead to changes in the functioning of the gut–brain axis, ultimately leading to mental disorders such as anxiety or depression. Gut microbiota also influences the hypothalamic–pituitary–adrenal (HPA) axis by modulating the secretion of neurotransmitters, which also leads to changes in brain function [8,9,10,11,12].

The aim of this review is to present current knowledge on the differences in microbiota structure in patients diagnosed with anxiety disorders, to discuss the potential mutual relationships between the symptoms presented by patients with anxiety disorders and quantitative differences in microbiota, as well as to highlight additional therapeutic opportunities associated with the use of probiotics.

## 2. Materials and Methods

This review is based on an analysis of the available literature concerning the relationship between gut microbiota and symptoms of anxiety disorders, with a particular focus on the potential therapeutic role of probiotics. A literature search was conducted using PubMed, Embase, CINAHL, and Google Scholar, considering full-text articles published in English, without any time restrictions. The review includes both clinical studies and randomized controlled trials involving individuals diagnosed with anxiety disorders.

## 3. Results

### 3.1. Dysbiosis and Anxiety Disorders

Hai-yin Jiang and colleagues conducted a study on the gut microbiome in patients with generalized anxiety disorder (GAD). They found that 40 patients with GAD, compared to the control group composed of 12 patients, exhibited lower diversity and richness of microorganisms, as well as a characteristic metagenomic composition, with reduced levels of bacteria that produce short-chain fatty acids and an increased abundance of bacteria such as *Escherichia-Shigella*, *Fusobacterium*, and *Ruminococcus gnavus*. Importantly, these changes in the microbiome persisted even during GAD remission, suggesting that the gut microbiome may influence the development of GAD and thus may be a potential therapeutic target for reducing the severity of anxiety symptoms [13].

Similarly, Chen Yi-huan et al. also demonstrated that the composition of the gut microbiome is altered in patients with generalized anxiety disorder. The levels of 25 bacterial genera significantly differed between the groups of affected individuals and the control group. The amounts of *Eubacterium coprostanoligenes*, *Ruminococcaceae UCG-014*, and *Prevotella 9* were more abundant in the control group and showed a negative correlation with anxiety severity and a positive correlation with its reduction. Conversely, *Bacteroides* and *Escherichia-Shigella* were more frequently represented in patients with anxiety disorders and had a positive association with anxiety severity. Based on these observations, the authors suggested that the altered gut microbiome profile plays a role in the pathogenesis and remission of generalized anxiety disorder [14].

Similar conclusions indicating the role of the microbiome in the development of emotional disorders were reached by Chinese scientists led by Jianbing Li, who investigated the relationship between gut microbiota and anxiety disorders (AD) using the two-sample Mendelian randomization (MR) method. They identified nine bacterial traits at the genus level that were significantly associated with the risk of AD. Four of these, such as *Blautia* and *Butyricicoccus*, showed a negative correlation with AD, suggesting that their higher presence may reduce the risk of these disorders. Five other genera, including *Eubacterium* brachy and *Coprococcus*, positively correlated with AD, indicating that their greater abundance may increase the risk of developing these disorders. The research methods used by the scientists included the weighted median and MR–Egger method, whose results confirmed those obtained by the inverse variance weighted (IVW) method. These studies underscore the role of gut microbiota in the pathogenesis of generalized anxiety disorder and suggest novel therapeutic strategies [15]. Cheng Yuanyuan and his team also conducted a study aimed at examining the relationship between the development of anxiety disorders and gut flora. In addition to analyzing the correlation between gut flora structure and anxiety disorders, the study also included an analysis of inflammation and polymorphisms of the brain-derived neurotrophic factor (BDNF) gene. The study included 48 individuals diagnosed with generalized anxiety disorder and 57 healthy participants as a control group. Differences in BDNF gene polymorphisms, gut microbiota distribution, and inflammatory and immunological markers were analyzed. It was found that the levels of TNF-alpha, interleukins 4 and 10, as well as IgG, were higher in patients with anxiety disorders. In the microbiota analysis, it was found that the strains *Paraprevotella*, *Euryarchaeota*, Caldivirga, *Porphyromonadaceae*, and *Desulfovibrionales* were more numerous in individuals with anxiety disorders, while the strains *Lactobacillus*, *Vagococcus*, *Barnesiella*, and *Paludibacter* were more common in healthy individuals. This study highlights the complex relationships between gut microbiota, inflammatory and genetic factors, and the development of anxiety disorders [16].

### 3.2. Dysbiosis and Anxiety and Depressive Disorders

Recent studies suggest that gut microorganisms, particularly bacteria from the *Firmicutes* and *Bacteroidetes* groups, can influence mental health through the gut–brain axis. Dysbiosis, or an imbalance in the gut microbiota, has been linked to anxiety, depression, and other mental health issues. At the same time, dietary components such as probiotics (*Lactobacillus*, *Bifidobacterium*), prebiotics (dietary fiber, alpha-lactalbumin), synbiotics, postbiotics (short-chain fatty acids), dairy products, spices (e.g., *Zanthoxylum bungeanum*, curcumin, capsaicin), fruits, vegetables, and medicinal herbs may exhibit protective effects against the development of psychopathological symptoms. They support the growth of beneficial gut microbiota while inhibiting the growth of harmful microorganisms. The use of these substances and products can be applied both in the prevention and alleviation of existing emotional disorder symptoms by promoting appropriate dietary habits [17].

Based on the gut–brain axis theory, researchers decided to develop an innovative approach to the differential diagnosis of depressive disorders and generalized anxiety disorder. The study included 54 participants: 23 patients with major depressive episodes, 21 patients with generalized anxiety disorder, and 10 healthy individuals as a control group. The microbiota analysis showed that in individuals with major depression, the abundance of *Sutterella* and *Fusicatenibacter* bacteria was significantly lower compared to the control group, while those with anxiety disorders had lower levels of *Fusicatenibacter* and *Christensenellaceae*. Additionally, in the group with anxiety disorders, a higher abundance of *Sutterella* and lower *Faecalibacterium* was observed compared to those diagnosed with major depression. A negative correlation was noted between the abundance of *Christensenellaceae* types and scores on the Hamilton Depression Rating Scale (HAMD). The study’s authors emphasize that gut microbiota could serve as a molecular marker for distinguishing between depressive and anxiety disorders [18].

The aim of another study was to identify biomarkers based on gut microbiota that could assess and determine the severity of depressive symptoms. The gut microbiota of 45 patients with untreated, severe depressive episodes was analyzed. Analysis of the 17-item Hamilton Depression Rating Scale and the 14-item Hamilton Anxiety Rating Scale indicated a correlation between microbiota alterations and symptom severity. Linear regression analysis revealed that higher scores on the Hamilton Depression Rating Scale were positively correlated with the presence of the genera *Akkermansia*, *Phascolarctobacterium*, *Coprococcus*, and *Streptococcus*, and negatively with *Clostridium* and *Bacteroides fragilis*. An increase in *Streptococcus* abundance was particularly useful for distinguishing between different levels of symptom severity. Furthermore, microbiome metabolite analysis suggested that indole-3-carboxyaldehyde could serve as a potential marker for differentiating the severity of depressive symptoms [19].

### 3.3. Dysbiosis and Anxiety Disorders with Co-Occurring Somatic Conditions

In 2022, a study was conducted to investigate how changes in the gut microbiome are associated with psychological factors, particularly in the context of generalized anxiety disorder. The study included patients divided into several groups: 35 patients with functional gastroenteropathy and generalized anxiety disorder, 16 patients with functional gastroenteropathy without anxiety disorders, 44 patients without functional gastroenteropathy but with diagnosed generalized anxiety disorder, and 30 healthy individuals as a control group. Stool samples were collected from all participants, and psychological assessments were conducted using questionnaires (e.g., Hamilton Anxiety Scale, Hamilton Depression Scale, Neuroticism Personality Questionnaire, Illness Perception Questionnaire, Toronto Alexithymia Scale, Symptom Severity Scale, and Cognitive Emotion Regulation Questionnaire) to evaluate the mental state of the studied groups. Based on the results, the researchers concluded that the gut microbiota of patients across all groups was relatively similar; however, the abundance of *Haemophilus influenzae* was significantly higher in patients with active gastrointestinal disease without additional anxiety disorders. In contrast, the group with functional gastrointestinal disease and anxiety disorders showed an increased relative abundance of microorganisms from the *Clostridium* group. Additionally, an increased abundance of *Fusobacterium* and *Megamonas* was observed in individuals with affective difficulties in describing feelings and difficulties in identifying emotions [20].

### 3.4. Mechanisms Linking Gut Microbiota and Anxiety Disorders

Scientific research has increasingly highlighted the crucial role of gut microbiota in mental health, particularly in relation to anxiety disorders. Several biological mechanisms underpin the connection between bacterial presence/absence in the gut and anxiety symptoms:A.Gut–Brain Axis and Neurotransmitter Production

The gut microbiota and the central nervous system are closely connected through the gut–brain axis. This communication occurs via the vagus nerve, immune signaling, and microbial byproducts. Certain bacteria, including *Lactobacillus* and *Bifidobacterium*, contribute to the production of gamma-aminobutyric acid (GABA), an essential neurotransmitter that helps regulate anxiety and stress responses. Reduction in these beneficial bacteria can be linked to heightened anxiety symptoms.

B.Inflammation and Immune System Modulation

An imbalance in gut microbiota, known as dysbiosis, can compromise the integrity of the intestinal barrier, leading to increased permeability. This allows pro-inflammatory molecules to enter the bloodstream, which can contribute to systemic inflammation. Chronic inflammation has been associated with heightened anxiety levels and changes in brain function. Research indicates that individuals with generalized anxiety disorder tend to have higher levels of inflammatory markers, which may stem from alterations in gut microbiota composition

C.Short-Chain Fatty Acids (SCFAs) and Metabolic Influence

Some beneficial gut bacteria, such as *Faecalibacterium prausnitzii* and *Blautia*, produce short-chain fatty acids (SCFAs) like butyrate, acetate, and propionate. These compounds play an important role in reducing neuroinflammation and influencing neurotransmitter activity.

D.Hypothalamic–Pituitary–Adrenal (HPA) Axis Regulation

The gut microbiota also plays a role in regulating the hypothalamic–pituitary–adrenal (HPA) axis, which controls the body’s response to stress. When gut bacteria are imbalanced, the HPA axis may become overactive, leading to elevated cortisol levels and increased anxiety symptoms. Some research suggests that probiotic supplementation, particularly with Lactobacillus rhamnosus, may help regulate stress responses and reduce HPA axis hyperactivity

### 3.5. Probiotics and Their Impact on the Intensity of Anxiety Symptoms

A.
*Lactobacillus*


Ruizhe Zhu et al. investigated the potential impact of *Lactobacillus plantarum JYLP-326*, a probiotic strain originally isolated from fermented dairy products, on the mental state of students experiencing anxiety and stress-related disorders. The study involved sixty participants who were randomly assigned to a placebo or probiotic group. The probiotic group received 1 × 10⁹ CFU per dose, administered twice daily, for a total of three weeks. Mental state assessments were conducted using the Athens Insomnia Scale (AIS-8), Hamilton Depression Rating Scale (HDRS-17), and Hamilton Anxiety Rating Scale (HAMA-14) at baseline and after the intervention. 16S rRNA sequencing and untargeted metabolomics were performed to analyze changes in gut microbiota and fecal metabolism. The results indicated that *JYLP-326* supplementation significantly alleviated symptoms of anxiety, depression, and insomnia, potentially by modulating gut microbiota composition and metabolism [21].

In another study researchers performed a randomized, double-blind, placebo-controlled trial that assessed the impact *of Lactobacillus plantarum P8*—a probiotic strain isolated from fermented food sources—on stress, anxiety, memory, and cognitive functions in 103 adults with an average age of 31.7 ± 11.1 years, all of whom exhibited moderate stress levels according to the Perceived Stress Scale (PSS-10). Participants consumed 10⁹ CFU per day for four weeks, with stress levels assessed using the PSS-10 and the Depression, Anxiety, and Stress Scale (DASS-42). The probiotic group demonstrated significant reductions in stress (mean difference 2.94; 95% CI 0.08 to 5.73; *p* = 0.048) and anxiety (mean difference 2.82; 95% CI 0.35 to 5.30; *p* = 0.031) compared to the placebo group. Additionally, the overall DASS-42 scores were significantly lower (mean difference 8.04; 95% CI 0.73 to 15.30; *p* = 0.041). Furthermore, inflammatory cytokines IFN-γ (mean difference 8.07 pg/mL; 95% CI −11.2 to −4.93; *p* < 0.001) and TNF-α (mean difference 1.52 pg/mL; 95% CI −2.14 to −0.89; *p* < 0.001) were significantly reduced, suggesting an anti-inflammatory mechanism contributing to symptom improvement. While differences in plasma cortisol levels between the groups were minor (mean difference 3.28 ug/dl; 95% CI −7.09 to 0.52; *p* = 0.090), the probiotic group exhibited notable improvements in verbal memory and cognitive functions, with effects varying between men and women. These findings indicate that *Lactobacillus plantarum P8* is an effective and natural remedy that not only reduces stress and anxiety symptoms but also enhances cognitive performance in stressed adults [22].

A study conducted by Wauters et al. examined the effects of *Lactobacillus rhamnosus CNCM I-3690*, a strain isolated from human gut microbiota, on academic stress in 46 healthy students. Participants were randomly assigned to receive acidified milk containing 10⁹ CFU of *Lactobacillus rhamnosus* twice daily for four weeks, while the control group received placebo. Psychological assessments, including the State-Trait Anxiety Inventory (STAI) and salivary cortisol measurements, revealed that the probiotic group exhibited a significantly lower increase in stress-induced anxiety compared to placebo, despite no observed changes in gut barrier permeability [23].

In contrast to the above-mentioned study, other researchers did not observe significant differences between the group supplementing *L. rhamnosus* and the control group [24].

The effect of probiotics on the physiological symptoms of anxiety in professional soccer players, who are exposed to stress due to intense training and competitions, may have a beneficial impact on brain function and, in turn, on their mental health. In a randomized, double-blind, placebo-controlled trial, 20 soccer players participated. For eight weeks, one group received probiotics (*Lactobacillus Casei Shirota*, 3 × 10^10^ CFU), while the other received a placebo. The results were analyzed using ANOVA and showed that after eight weeks, there were no significant differences between the groups in heart rate and electrodermal responses. However, significant changes were observed in theta and delta brain waves after four weeks and improved reaction times in a cognitive test in the probiotic group [25].

An unusual topic was explored by Rahim Badrfam et al., who studied the effect of the probiotic *Lactobacillus acidophilus* on psychopathological symptoms in sixty methamphetamine-addicted patients with psychotic symptoms. The results showed that patients taking probiotics reported better sleep quality and appetite and presented higher body mass index compared to the placebo group. However, no significant improvement in psychotic and anxiety symptoms was observed between the groups. The authors suggest that probiotics may positively affect sleep quality and appetite in patients with chronic methamphetamine addiction, although they do not significantly impact psychotic symptoms [26].

The effectiveness of another species of *Lactobacillus* was studied by Kensei Nishida et al., who evaluated the effect of daily consumption of the probiotic *Lactobacillus gasseri CP2305* on alleviating stress symptoms in sixty medical students preparing for a state exam. The results showed that CP2305 supplementation significantly reduced anxiety and sleep disturbances compared to placebo, as assessed by the Pittsburgh Sleep Quality Index and the Spielberger State-Trait Anxiety Inventory. Additionally, electroencephalogram analysis showed that the probiotic shortened sleep onset time and reduced the duration of night awakenings. Stool sample analysis revealed that CP2305 prevented stress-induced decrease in *Bifidobacterium* spp. and increase in *Streptococcus* spp. [27].

B.
*Bifidobacterium*


Michael P. Siegel and collaborators conducted a week-long study to determine if *Bifidobacterium longum* could reduce levels of stress, anxiety, and depression symptoms. The study involved 84 participants who were randomly assigned to either a placebo or probiotic group. The researchers measured stress (PSS), depression (CES-D-Center for Epidemiologic Studies Depression Scale), and anxiety (STAI) levels before and after the week-long probiotic intervention. The results did not show a significant reduction in stress, depression, or anxiety. The authors suggested that the lack of effect might be due to the use of a single probiotic strain and the short duration of the study, which could have limited its effectiveness [28].

A similar study using *Bifidobacterium longum* was conducted by Marcus Boehme et al. The research team’s results showed that supplementation significantly reduced perceived stress and improved sleep quality in the treatment group which included 47 patients compared to the placebo. Despite the absence of significant changes in salivary cortisol levels, there was a trend toward reduced stress response in the probiotic group. Stress reduction following the intervention was also correlated with a decrease in anxiety and depression symptoms. Both studies provide valuable insights into the potential benefits of probiotics in reducing stress levels and the intensity of anxiety and depression symptoms, though effectiveness may depend on factors such as the duration of the intervention, the number of strains used, and other variables [29].

Okubo R. and the team studied the effects of the probiotic *Bifidobacterium breve A-1* on anxiety and depressive symptoms in patients with schizophrenia, as well as its impact on the immune system. In an open-label study, all participants took *B. breve* A-1 for 4 weeks, followed by a 4-week observation period. Changes were assessed using the Hospital Anxiety and Depression Scale (HADS) and the Positive and Negative Syndrome Scale (PANSS). After 4 weeks, significant improvements were observed in HADS and PANSS scores. In 12 out of 29 patients, HADS scores were reduced by more than 25%, indicating a positive response to the treatment. Patients who responded well to the treatment had fewer negative symptoms, higher levels of *Parabacteroides* bacteria in their gut, and consumed fewer dairy products. An increase in the expression of IL-22 and TRANCE proteins was also observed in these patients. The study suggests that B. breve A-1 may improve anxiety and depressive symptoms in patients with schizophrenia [30].

C.Multicomponent Probiotic Formulations

Michael Messaoudi and his team conducted a study on the anxiolytic effects of the probiotics *Lactobacillus Helveticus R0052* and *Bifidobacterium Longum R0175*. The study consisted of two parts: an experiment on rats and a clinical trial on sixty-six healthy volunteers. In the clinical part, participants took probiotics for 30 days in a double-blind trial. Anxiety, depression, and stress levels were assessed using various scales, including HSCL-90 and HADS, and urinary cortisol levels were analyzed. The results showed that daily use of probiotics provided psychological benefits, improving scores on the HSCL-90 and HADS scales [31].

A different approach was taken by Nhan Tran et al., who analyzed the effects of probiotics on anxiety and other related changes, such as anxiety control and mood regulation. The study involved 86 students in a double-blind trial, randomly assigning them to four probiotic groups or one placebo group. The groups varied in CFU (colony formation unit) levels and bacterial diversity. Anxiety was assessed using various questionnaires. The results showed that probiotics could significantly reduce anxiety, particularly in groups with high CFU levels, regardless of bacterial diversity. Women showed greater reductions in negative emotions, while men exhibited greater reductions in autonomic anxiety. Participants of African/African American descent had the greatest improvements, suggesting that gender and ethnicity may influence the effectiveness of probiotics [32].

However, the results from Sevda Eskandarzadeh et al. did not confirm previous conclusions made by other authors. The researchers aimed to evaluate the effectiveness of probiotics as an adjunct therapy in patients with generalized anxiety disorder (GAD). The study involved 48 patients diagnosed with GAD who had not used pharmacotherapy before the study. Participants were randomly divided into two groups: one took probiotics along with 25 mg of sertraline, and the other took sertraline with a placebo for 8 weeks. The probiotics contained strains of *Bifidobacterium bifidum*, *Bifidobacterium lactis*, *Bifidobacterium longum*, and *Lactobacillus acidophilus*. The primary goal was to reduce the severity of anxiety, assessed using the HAM-A scale, as well as BAI, STAI, and WHOQOL-BREF to evaluate quality of life. The results showed that after 8 weeks, the group combining probiotics with sertraline had significantly lower scores on the HAM-A scale, indicating a reduction in anxiety. The BAI scale also showed a greater reduction in anxiety than in the placebo group, although the difference was not statistically significant. Despite greater improvement in the State-Anxiety Inventory scores in the probiotic group, the Trait-Anxiety Inventory scores did not show significant differences between the groups. The quality-of-life analysis (WHOQOL-BREF) did not show significant changes in either group after the study [33].

## 4. Summary

Our review highlights the important role of gut microbiota in the pathogenesis and progression of anxiety and depressive disorders. The results of the studies suggest that dysbiosis is often observed in patients with various forms of anxiety disorders. Significant changes in gut microbiota composition are strongly associated with anxiety symptoms. Studies indicate that individuals with anxiety disorders exhibit a reduced abundance of beneficial bacteria such as *Lactobacillus*, *Bifidobacterium*, *Faecalibacterium prausnitzii*, and Blautia, which play key roles in short-chain fatty acid (SCFA) production, neurotransmitter modulation, and anti-inflammatory responses. Conversely, an overgrowth of pathogenic or pro-inflammatory bacteria, including *Escherichia-Shigella*, *Fusobacterium*, and *Ruminococcus gnavus*, has been linked to increased gut permeability and heightened systemic inflammation, both of which contribute to the severity of anxiety symptoms [14,16].

Probiotic interventions targeting the restoration of microbial balance have shown promise in reducing anxiety symptoms, with particular effectiveness observed in strains such as *Lactobacillus plantarum*, *Bifidobacterium longum*, and *Lactobacillus rhamnosus* [22,23]. However, further research is required to optimize strain selection, dosage, and treatment duration for maximum therapeutic benefit.

Probiotics, which modulate the gut microbiota composition, may serve as an important therapeutic tool in the treatment of anxiety disorders. Research showed that probiotic supplementation can improve sleep quality, appetite, and overall well-being of patients, although it does not always directly affect the reduction of anxiety symptoms. Additionally, some studies demonstrated the potential use of gut microbiota as a biomarker for differential diagnosis of anxiety and depressive disorders, which could enable a more targeted approach to treatment.

In summary, gut microbiota plays a crucial role in mental health and represents an important target in anxiety disorder therapy. Further research is needed to better understand the mechanisms through which gut microbiota influences brain function and to develop effective therapeutic interventions based on microbiome modulation. Future research should explore multi-strain probiotic combinations, personalized gut microbiota-targeted treatments, and the interaction between dietary modifications and microbiota composition in anxiety disorders.

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
