# Peer review of "The Impact of Gut Microbiota on the Development of Anxiety Symptoms—A Narrative Review"

_nutrients, 2025, doi:10.3390/nu17060933_

Round 1

Reviewer 1 Report

Comments and Suggestions for Authors

This review explores alterations in the gut microbiota of individuals with anxiety disorders and examines the potential therapeutic effects of probiotics. The authors concluded that gut dysbiosis appears to be closely linked to anxiety disorders, and probiotic interventions could represent a promising therapeutic avenue.

Some suggestions/comments:

1.Abstract, lines 15-17, delete please “ Background/Objectives: A  few sentences to place the question addressed in a broader context and highlight the purpose of the study”.

  1. Introduction:

-lines 38-39: add please which are the “Neurotransmitter precursors”.

-lines 43-44, you wrote “anxiety disorders are among the most common mental health  problems, affecting 4.09% of the population”. In which year was this statistic made and where? Please add.

  1. Results:

-only for a part of the studies presented you added the number of patients involved in the study. For uniformity, in my opinion you should add the number of patients included in the study for all the studies.

  1. pages 4-7, Probiotics and Their Impact on the Intensity of Anxiety Symptoms:

- for all the presented studies add please the origin of the probiotics, the amount of the probiotic administrated and the duration of administration

-page 6, point C. Multicomponent Probiotic Formulations: describe please the study performed on rats.

  1. page 2, lines 66-68, you wrote: “The review includes both clinical studies on animal models and randomized controlled trials involving individuals diagnosed with anxiety disorders”.

In your review, no animal studies are presented.

6. In the review are presented studies taken from the literature. The scientific explanations concerning the connection between bacteria presence/absence and anxiety are missing. 7. Summary, lines 331-334, you wrote “Significant changes in the microbiota composition, including reduced diversity and specific alterations in the abundance of certain bacteria, such as increased levels of Escherichia-Shigella and decreased levels of short-chain fatty acid-producing bacteria, are associated with the severity of anxiety symptoms”. In my opinion you must give more details about the “good/bad” bacteria related to anxiety.

8. As you wrote in the introduction “The microbiota is composed primarily of bacteria, but also includes viruses, fungi, and archaea, which interact with each other and with the host organism”. Please specify if there is a connection between viruses, fungi, archaea and anxiety. 

9. The number of references is too small for a review article.

Author Response

Dear Editor,  

I would like to sincerely thank you for the time and effort you dedicated to reviewing our work. Your feedback has been invaluable and allowed us to view our work from a new perspective. We are truly grateful for your constructive comments and suggestions, which will help us refine the final version.

Thank you for your professionalism, patience, and dedication. Your assistance has been invaluable, and we hope we will have the opportunity to collaborate again in the future.

We have referred to your suggestions in red colour.

Best regards,
Authors

  1. Abstract, lines 15-17, delete please “ Background/Objectives: A  few sentences to place the question addressed in a broader context and highlight the purpose of the study”.

Thank you for your suggestion. The phrase in question has been removed, we sincerely apologize for the oversight, and have entered the correct purpose of the work.

  1. Introduction:

-lines 38-39: add please which are the “Neurotransmitter precursors”.

-lines 43-44, you wrote “anxiety disorders are among the most common mental health  problems, affecting 4.09% of the population”. In which year was this statistic made and where? Please add.

Thank you for your suggestion. We have updated the list of neurotransmitters affected by the microbiota.

We have also revised the statistical data regarding the statement that “anxiety disorders are among the most common mental health problems.” The source of this information is reference 6 in the bibliography, a publication from 2023. In parentheses, after the percentage value, we have added the year from which the data originate.

Syed Fahad Javaid, et al. “Epidemiology of Anxiety Disorders: Global Burden and Sociodemographic Associations.” Middle East 400Current Psychiatry, vol. 30, no. 1, 26 May 2023, mecp.springeropen.com/articles/10.1186/s43045-023-00315-3, 401 https://doi.org/10.1186/s43045-023-00315-3.

  1. Results:

-only for a part of the studies presented you added the number of patients involved in the study. For uniformity, in my opinion you should add the number of patients included in the study for all the studies.

Thank you for your suggestions, the corrections have been made.

  1. pages 4-7, Probiotics and Their Impact on the Intensity of Anxiety Symptoms:

- for all the presented studies add please the origin of the probiotics, the amount of the probiotic administrated and the duration of administration

For the section “Probiotics and Their Impact on the Intensity of Anxiety Symptoms,” we have provided information on the origin of probiotics, the administered dosage, and the duration of administration.

-page 6, point C. Multicomponent Probiotic Formulations: describe please the study performed on rats.        

The study conducted on rats was a pilot study that was later carried out on patients. The study on patients is described in the same section of the text.

  1. page 2, lines 66-68, you wrote: “The review includes both clinical studies on animal models and randomized controlled trials involving individuals diagnosed with anxiety disorders”.

In your review, no animal studies are presented.

In your review, no animal studies are presented. Thank you for pointing out the error, the sentence has been corrected.

  1. In the review are presented studies taken from the literature. The scientific explanations concerning the connection between bacteria presence/absence and anxiety are missing. 

We have added explanations in the form of a new section titled “Dysbiosis and Anxiety Disorders,” which discusses the relationship between the presence or absence of certain bacteria and anxiety disorders. In this section, we have included biological mechanisms such as the impact of gut microbiota on the gut-brain axis, neurotransmitter production, and immune system modulation.

  1. Summary, lines 331-334, you wrote “Significant changes in the microbiota composition, including reduced diversity and specific alterations in the abundance of certain bacteria, such as increased levels of Escherichia-Shigella and decreased levels of short-chain fatty acid-producing bacteria, are associated with the severity of anxiety symptoms”. In my opinion you must give more details about the “good/bad” bacteria related to anxiety.

Thanks for your advice, we've edited and expanded the summary section to include your tips.

  1. As you wrote in the introduction “The microbiota is composed primarily of bacteria, but also includes viruses, fungi, and archaea, which interact with each other and with the host organism”. Please specify if there is a connection between viruses, fungi, archaea and anxiety. 

Thank you for your valuable advice, we have provided the importance of viruses, fungi and archaea in the context of the impact of microbiota on anxiety disorders.

  1. The number of references is too small for a review article.

Thank you for your valuable attention, we have added a fourth item to the bibliography to enrich our research.

Reviewer 2 Report

Comments and Suggestions for Authors

Journal: Nutrients (ISSN 2072-6643)

Manuscript ID: nutrients-3500658

Type: Review

Title: The Impact of Gut Microbiota on the Development of Anxiety Symptoms – a narrative review

This review titled “The Impact of Gut Microbiota on the Development of Anxiety Symptoms” by Kamil Nikel et al, study offers a well-structured and informative explanation of the gut-brain axis, neurotransmitter regulation, and immune system participation. The inclusion of preclinical and clinical data makes the review more complete and useful to both academics and physicians. However, some comments need to be address to increase the paper's clarity, depth, and scientific rigor.

1: Lines 62-68: The review claims that PubMed, Embase, CINAHL, and Google Scholar were utilized to conduct literature searches.
However, it does not provide inclusion/exclusion criteria (for example, publication date range, research type, or demographic characteristics).

2: Provide a detailed explanation of the mechanistic pathways involved in the gut-brain axis, immune responses, and neurotransmitter modulation?

3: The review does not discuss how genetics, nutrition, age, and geographical impact gut microbiota composition and its function in anxiety disorders.

4: Lines 177-324 Some research indicate that Lactobacillus plantarum, Lactobacillus rhamnosus, and Bifidobacterium breve alleviate anxiety, while others show no effect.
The review doesn't explain why these contradictions occur.

5: The most cited research lasted between 4 and 12 weeks, although the long-term effect of probiotics on anxiety symptoms is unclear.

6: Lines 71-89: The review contains both human and animal studies, however it does not address the translational value of animal models.

7: Lines 328-346 Future research should explore multi-strain probiotic combinations, personalized gut microbiota-targeted treatments, and the interaction between dietary modifications and microbiota composition in anxiety disorders.

8: Lines 43-44: Redundant phrasing: "These studies, like the previous ones, highlight the role of gut microbiota in the pathogenesis of generalized anxiety disorders and point to a new direction for treating these disorders."

(e.g., "highlight the role of gut microbiota" and "point to a new direction" both suggest the same idea).

9: Lines 79-81 Repetitive structure ("In contrast, JYLP-326 supplementation appeared..." and "In another study...").

 Slightly awkward transition between the two sentences.

10: Lines 40-41. "Five other researchers have confirmed this finding in recent years."

"Five other researchers" is vague—who are they?

"In recent years" is imprecise—how recent?

Past perfect ("have confirmed") is unnecessary since no prior event is being referenced.

Suggestion: "Multiple studies in the past decade have confirmed this finding."

11: Lines 13-14: "Since some species of gut bacteria can be pathogenic, it is important that those which positively impact intestinal processes and patient health predominate."

Those which" should be "those that." Overly wordy phrasing.

 Suggestion: "Since some gut bacteria are pathogenic, beneficial species should predominate to support intestinal health."

12: Lines 60-61: "Based on the results of the 17-item Hamilton Depression Rating Scale and the 14-item Hamilton Anxiety Scale, the authors concluded that microbiota changes correlated with symptom severity."

"Based on the results" suggests that the authors themselves were analyzed instead of the data.

"Microbiota changes correlated with symptom severity" is vague.

Suggestion: "Analysis of the 17-item Hamilton Depression Rating Scale and 14-item Hamilton Anxiety Scale indicated a correlation between microbiota alterations and symptom severity."

Author Response

Dear Editor,

We would like to sincerely thank you for the time and effort you dedicated to reviewing our work. Your feedback has been invaluable and allowed us to view our work from a new perspective. We are truly grateful for your constructive comments and suggestions, which will help us refine the final version.

Thank you for your professionalism, patience, and dedication. Your assistance has been invaluable, and we hope we will have the opportunity to collaborate again in the future.

We have referred to your suggestions in purple colour.

Best regards,
Authors

1: Lines 62-68: The review claims that PubMed, Embase, CINAHL, and Google Scholar were utilized to conduct literature searches.
However, it does not provide inclusion/exclusion criteria (for example, publication date range, research type, or demographic characteristics).

We have used keywords that appropriately describe the topic of the paper in databases. Since this is a narrative review, keywords are not required. However, if you would like us to adjust them, please let us know.

2: Provide a detailed explanation of the mechanistic pathways involved in the gut-brain axis, immune responses, and neurotransmitter modulation?

We have added an entire section titled “Mechanisms Linking Gut Microbiota and Anxiety Disorders,” which includes all the mechanisms through which the gut microbiota influences anxiety disorders.

3: The review does not discuss how genetics, nutrition, age, and geographical impact gut microbiota composition and its function in anxiety disorders.

In the introduction, we have added a section describing the impact of genetics, nutrition, age, and geographical factors on gut microbiota composition. Thank you for your valuable suggestion.

4: Lines 177-324 Some research indicate that Lactobacillus plantarum, Lactobacillus rhamnosus, and Bifidobacterium breve alleviate anxiety, while others show no effect.
The review doesn't explain why these contradictions occur.

Explanations of the effects of gut microbiota components on anxiety disorders have been included in the newly added section “Mechanisms Linking Gut Microbiota and Anxiety Disorders.”

5: The most cited research lasted between 4 and 12 weeks, although the long-term effect of probiotics on anxiety symptoms is unclear.

Thank you for pointing this out. In the conclusion, we have included information stating that further research is needed to assess the long-term effects.

6: Lines 71-89: The review contains both human and animal studies, however it does not address the translational value of animal models.

The study conducted on rats was a pilot study, which was later replicated in a study involving patients. The patient study is described in the same section of the text.

7: Lines 328-346 Future research should explore multi-strain probiotic combinations, personalized gut microbiota-targeted treatments, and the interaction between dietary modifications and microbiota composition in anxiety disorders.

We have implemented your suggestion, and the summary section has been revised accordingly. Thank you for your advice.

8: Lines 43-44: Redundant phrasing: "These studies, like the previous ones, highlight the role of gut microbiota in the pathogenesis of generalized anxiety disorders and point to a new direction for treating these disorders."

(e.g., "highlight the role of gut microbiota" and "point to a new direction" both suggest the same idea).

Thank you for your advice. We have reworded the sentence according to your suggestions: “These studies highlight the role of gut microbiota in the pathogenesis of generalized anxiety disorders and propose novel therapeutic strategies.”

9: Lines 79-81 Repetitive structure ("In contrast, JYLP-326 supplementation appeared..." and "In another study...").

 Slightly awkward transition between the two sentences.

We sincerely thank you for your valuable advice. Sections A, B, and C of the “Probiotics and Their Impact on the Intensity of Anxiety Symptoms” have been revised in accordance with the suggestions of both reviewers.

10: Lines 40-41. "Five other researchers have confirmed this finding in recent years."

"Five other researchers" is vague—who are they?

"In recent years" is imprecise—how recent?

Past perfect ("have confirmed") is unnecessary since no prior event is being referenced.

Suggestion: "Multiple studies in the past decade have confirmed this finding."

We sincerely appreciate your suggestion; however, we would like to clarify that such phrases are not present in the text. The authors are unsure about the proposed change and would appreciate further clarification.

11: Lines 13-14: "Since some species of gut bacteria can be pathogenic, it is important that those which positively impact intestinal processes and patient health predominate."

Those which" should be "those that." Overly wordy phrasing.

 Suggestion: "Since some gut bacteria are pathogenic, beneficial species should predominate to support intestinal health."

Thank you for your valuable advice. The change has been made in accordance with your suggestion.

12: Lines 60-61: "Based on the results of the 17-item Hamilton Depression Rating Scale and the 14-item Hamilton Anxiety Scale, the authors concluded that microbiota changes correlated with symptom severity."

"Based on the results" suggests that the authors themselves were analyzed instead of the data.

"Microbiota changes correlated with symptom severity" is vague.

Suggestion: "Analysis of the 17-item Hamilton Depression Rating Scale and 14-item Hamilton Anxiety Scale indicated a correlation between microbiota alterations and symptom severity."

Thank you for your valuable suggestion. We have revised the sentence to make it clearer and more precise, as per your recommendation. Specifically, we changed “Based on the results” to “Analysis of,” and rephrased the part regarding microbiota changes to avoid vagueness. We hope this revision better aligns with your expectations.

Round 2

Reviewer 1 Report

Comments and Suggestions for Authors

In my opinion all the corrections are well done and the article can be published in the present form.

Best regards,

Author Response

Dear Reviewer,

We sincerely thank you for your time and for the previously suggested improvements. Your contribution has enhanced the value of our work for the readers.

With respect,
The Authors